# The Effect of High-Intensity Interval Training Periods on Morning Serum Testosterone and Cortisol Levels and Physical Fitness in Men Aged 35–40 Years

**DOI:** 10.3390/jcm10102143

**Published:** 2021-05-15

**Authors:** Tadeusz Ambroży, Łukasz Rydzik, Zbigniew Obmiński, Wiesław Błach, Natalia Serafin, Blanka Błach, Jarosław Jaszczur-Nowicki, Mariusz Ozimek

**Affiliations:** 1Institute of Sports Sciences, University of Physical Education, 31-571 Krakow, Poland; tadek@ambrozy.pl (T.A.); mariusz.ozimek@awf.krakow.pl (M.O.); 2Department of Endocrinology, Institute of Sport-National Research Institute, 01-982 Warsaw, Poland; zbigniew.obminski@insp.pl; 3Faculty of Physical Education & Sport, University School of Physical Education, 51-612 Wroclaw, Poland; wieslaw.judo@wp.pl (W.B.); blanka.blach@awf.wroc.pl (B.B.); 4Faculty of Physical Education and Sport, Institute of Social Sciences, University of Physical Education in Krakow, 31-571 Kraków, Poland; natalia.ambrozy@gmail.com; 5Department of Tourism, Recreation and Ecology, University of Warmia and Mazury in Olsztyn, 10-719 Olsztyn, Poland; j.jaszczur-nowicki@uwm.edu.pl

**Keywords:** men, training period, blood, hormones, physical performance

## Abstract

Background: Intensive physical activity largely modulates resting concentrations of blood cortisol (C) and testosterone (T) and their molar ratio, which is defined as the anabolic–catabolic index and expressed as T/C × 10^2^. The aim of the study is to evaluate the effect of the author’s high-intensity training program on T, C, T/C × 10^2^, and selected physical fitness indices in men between 35 and 40 years of age. Methods: The experiment was conducted on a group of 30 healthy men, divided into control and experimental groups. The experimental group followed a high-intensity 8-week training program, which included three sessions per week, each of them lasting 1 h and consisting of intensive-interval exercises followed by strength circuit exercises. The controls did not change their previous recreational physical activity. T, C, and T/C × 10^2^ were measured before and after the experiment for all participants. Physical performance was examined using a standardized laboratory exercise test to determine maximal oxygen uptake (VO_2max_). Results: There were statistically significant increases in T (by 36.7%) and T/C × 10^2^ (by 59%), while C somewhat dropped (by 12%) in the experimental group. No changes in the hormonal indices were found in the control group. After completing the experimental training, there were no statistically significant changes in aerobic capacity, but it improved muscle strength in the men studied. Conclusions: High-intensity interval training, continued over an 8-week period, modulates (significantly and positively) the balance between testosterone and cortisol levels and improves physical capacity in men aged 35–40 years.

## 1. Introduction

As people age, adverse physical and physiological changes occur in the human body. Maximum skeletal muscle strength, mass, and endurance decrease as a consequence of morphological and physiological changes in the cells of the organs responsible for the production of important metabolic regulators and target tissues. There are several markers of these changes that are identifiable in blood and help distinguish biological age from chronological age [1]. Among these markers are steroid hormones belonging to the group of androgens, mainly testosterone. In men, the process of a gradual decrease in testosterone (T) synthesis begins at around 35–40 years of age [2,3]. In this age range, testosterone levels are reported to decrease by an average of 1.6% each year [4]. Testosterone plays an important role in regulating the metabolism, reproductive system, and mental status of men. A significant decrease in blood testosterone (<7.0 nM) is considered hypogonadism, which results in poor somatic and mental health. It should be emphasized that lifestyles and healthy behavior have a substantial effect on the androgenic status in men aged 30–40 years [5], and, as a consequence, it affects the general state of health [6,7].

Studies relating to the effect of training on testosterone levels have focused on the changes as a result of physical activity [8,9]. It should be noted, however, that if the training activity is too high, i.e., not adjusted to the biological capabilities of the human body, chronic fatigue syndrome (overtraining) may develop, leading to a decrease in testosterone production [10,11]. As mentioned earlier, some authors have suggested that circuit training alternated with high-intensity interval training may have a beneficial effect on general health, including muscular and skeletal systems [12]. Multiple observations have been made to determine the age- and exercise-dependent effects of a multi-week training period on changes in resting testosterone levels [13,14,15,16]. The results showed little or no change in testosterone levels, which is likely to have depended on the training modality. Apart from testosterone, the second most important hormone that regulates metabolism and exercise adaptations is cortisol (C). This hormone is produced by the adrenal cortex and is commonly called the stress hormone because its blood levels increase in response to strong stressors, including psychophysical stimuli (such as sports competitions) [12,17,18,19] and environmental stimuli (such as exposure to accelerations of >5G) [20] or sudden hypobaric hypoxia induced in laboratory settings (5000 m) [21]). As cortisol and testosterone have opposing effects on the rate of endogenous protein metabolism, the idea has emerged to assess the anabolic–catabolic balance as an index of the quotient of molar concentrations (T/C) [22,23]. A significant decrease in the T/C index indicates that the training activity is too high and catabolic processes prevail, which, in extreme cases, may lead to a decrease in skeletal muscle mass. It seems that men who undertake moderate physical activity at leisure to achieve better physical fitness and health status are not at such high risk. Nevertheless, amateur workouts of different structures, i.e., weekly training volume, session intensity, and the work-to-rest ratio during repeated exercises also cause different biological responses. An optimal training program for middle-aged men with inadequate physical activity levels should improve cardiorespiratory capacity, muscle strength, and metabolic status. Therefore, it seems appropriate to monitor their effects using exercise tests and hormone determinations.

The aim of the study is to evaluate the effect of the author’s high-intensity training program on testosterone, cortisol, and selected physical fitness indices in men between 35 and 40 years of age. The application value may be the verification of a training program that will favor the most effective hormone modulation.

## 2. Material and Methods

### 2.1. Study Group

The experiment conducted in this study involved a group of men (*n* = 30), aged 35 to 40 years, who were selected by purposive sampling, with the inclusion criterion being their sports skill level, determined before the experiment based on the results of a pilot study. Due to the high intensity of the training, only individuals who had been doing recreational training for at least 2 years were qualified to participate in the study. The participants were healthy and did not take any medication on a regular basis. The exclusion criteria were endocrine disorders and fertility problems. Detailed characterization of participants is presented in Table 1.

A pedagogical experiment was conducted in the study. The researchers conducted an intervention that involved manipulating the form and manner of physical activity of the participants. This was achieved by introducing an experimental training program in the form of high-intensity endurance and strength training into their daily physical activity. To conduct the experiment, the technique of working with two equal groups, formed from previously qualified participants, was used. The men were randomized into two groups: a control group and a study group (each consisting of 15 participants).

### 2.2. Research Program and Methodology

The first test for both subgroups was performed before the experiment. In the control group, the participants continued their previous form of activity, and, between the first and final examinations, they followed their previous program of recreational physical activity. In the study (experimental) group, a special training modification was introduced (independent variable), consisting of performing 60 min of personal endurance and strength training. The experiment lasted eight weeks. All training sessions were performed in the afternoon. Before and after this period, hormonal indices and physical capacity levels were examined (Figure 1).

Testosterone (T) and cortisol (C) levels and the anabolic–catabolic index, i.e., the molar T-to-C ratio, were the dependent variables that were measured in fasting venous blood collected in the morning (8:00 a.m.). The measurements were performed at the ALAB Laboratoria health center using the Roche test and the ECLIA (enhanced chemiluminescence immunoassay) method on the Cobas e601 system (Roche Diagnostics, Basel, Switzerland). The increase of maximal oxygen uptake (VO_2max_) during a graded exercise on a treadmill and the postexercise increase in the strength of selected muscle groups of the upper body and upper and lower limbs were adopted as measures of the improvement of physical capacity. Post-training changes in hormone indices and physical capacity, for each individual separately, were analyzed using Student’s *t*-test for dependent samples, whereas the test for independent samples was used to test intergroup comparisons before and after the experiment. The level of statistical significance was set at *p* < 0.05. Intraclass correlation coefficients (ICCs) and the coefficient of variation (CV%) were calculated. The magnitude of effect size for comparison between groups and conditions was expressed using Cohen’s d. When d ranges from 0 to 0.2, the effect is small, i.e., negligible; it is medium from 0.2 to 0.5, large from 0.5 to 0.8, and extremely large when over 1.4. Calculations were performed using STATISTICA software (ver. 13.3, StatSoft, Krakow, Poland). The participants in both groups did not change their diets during the experiment. Diets, sleep duration, and lifestyles were monitored by recording in notebooks and interviews.

Participants were informed of all research procedures prior to participation in the study, in accordance with the ethical principles of the WMA (The World Medical Association) Declaration of Helsinki (2000). The precondition for participation in the study was the participant’s written informed consent and a medical certificate of no contraindications to physical exercise. The experiment was approved by the Bioethics Committee at the Regional Medical Chamber (No. 309/KBL/OIL/2019).

### 2.3. Measurement of Physical Fitness

In this study, muscle strength andVO_2max_ were determined in the study groups using the following tests [24]:Evaluation of aerobic capacity. To assessVO_2max_, a running test with graded exercise intensity is performed on a treadmill (h/p/Cosmos, Nußdorf, Germany). The test begins with a 2-min recording of respiratory indices at rest, during which the participants remain in a standing position. During the first 4 min of the test, the participants run at a speed of 8 km·h^−1^. Next, the running speed is increased by 1 km·h^−1^ every 2 min. The effort is continued until the extreme fatigue of the participants, which is manifested by the inability to continue running at the set speed. During the test, the levels of cardiorespiratory indices are recorded based on the breath-by-breath method using an ergospirometer (Cosmed, Rome, Italy). The highest recorded value of minute oxygen uptake is considered to be VO_2max_ [25].Evaluation of abdominal strength (sit-ups). The tested person lies on the mattress with feet 30 cm apart and knees bent at a right angle. Hands are intertwined, resting on the neck. The participant is assisted by a partner who holds the participant’s feet so that they remain in contact with the ground. At the start signal, the participant sits up to touch their knees with their elbows and then returns to the starting position. The exercise duration is 30 s.Evaluation of shoulder girdle strength by the number of repetitions of pull-ups on a bar. The participant catches the bar with a pronated grip and hangs there; at the signal, the participant bends his arms at the elbow and pulls his body up so high that the chin is above the bar, and then, without a rest, returns to a simple hanging position. The exercise is repeated as many times as possible without rest; the result is the number of complete pull-ups (chin over the bar)Evaluation of the dynamic strength of lower limbs (long jump from a standing position). The participant stands with his feet slightly apart in front of the starting line and bends his knees and moves his arms backward at the same time; then, he performs an arm swing and jumps as far as he can. The landing occurs on both feet while maintaining the upright position. The test is performed twice.

### 2.4. Experimental Program

Implementation of the experimental program consisted of the introduction of the strength and endurance training designed for the experiment to the daily activity of the experimental group (*n* = 15). The authors made modifications to high-intensity interval training (HIIT) [26]. With its high diversity, the author’s program was supplemented with new methods and exercise structures considered to be most effective [27,28,29,30]. The main part of the training was based on different variants of circuit training and followed the principles of functional training [31,32,33]. Each training session was based on interval training according to the high-intensity interval training method and ended with strength circuit training (Table 2). The control group (Co) pursued their previous recreational physical activity, which was monitored but not programmed by the authors of this study.

To avoid fatigue and training monotony, an innovative training unit that was consistent with the entire program was planned for each session of the week. Each training session was preceded by a warm-up and included well-thought-out exercises, according to the author’s ideas. The training program was designed so that it was simple to perform and accessible to every participant. Individual exercises were performed at a fast pace (concentric phase: 1 s, eccentric phase: 2 s), with a particular focus on the correct technique of the movement tasks. Each study group trained three times a week using the assumed intensity and number of repetitions (Table 2).

## 3. Results

The reference range for T, determined in the laboratory, is 9.7–27.8 nmol/L. After the 8-week experiment, a statistically significant increase (36.7%) was observed in testosterone levels in the exercising group (Ex) that performed the HIIT training, while a 6% increase was not significant in the control group (Co). The relative intragroup variability of T concentrations expressed by CV% = (SD/X) × 100 (CV—coefficient of variation, SD—standard deviation) in the Ex group decreased after training (27.5% vs. 20.8%), whereas it remained unchanged in the Co group (30.6%). Neither baseline nor postexercise mean T concentrations differentiated statistically between the two groups, whereas absolute mean postexercise changes (ΔT) showed a significant intergroup difference. It is worth noting that the ICC (intraclass correlation coefficient) for T was statistically significant in both groups (0.826 for Ex and 0.880 for Co). Baseline T values that were below the lower limit of the reference range were found in one participant from the Ex group (8.1 nmol/L) and one from the Co group (8.0 nmol). Furthermore, two participants in the Co group had T < 9 nmol after the experiment (Table 3).

Physiological resting cortisol levels were within the range of 138–633 nM. Mean cortisol concentrations did not change significantly in both groups after the experiment. Before the training in the Ex group, C levels were significantly higher than in the Co group, but the differences disappeared after the experiment. This was due to a marked decrease in C (by 12%) in the Ex group and no changes in the Co group. The ICC for cortisol was significant (0.535) only in the Co group (Table 4).

The analysis of post-training changes in the anabolic–catabolic index revealed that training according to the HIIT program resulted in a significant increase in the anabolic–catabolic index (by 59%). Both the increase in T and the decrease in C in this group were responsible for such a large change. In the Co group, the 16% increase in the index was insignificant. The ICC for the index was significant in the Co group (0.690) but insignificant (0.274) in the Ex group (Table 5).

The results in Table 6 show the values of VO_2max_. The calculations showed statistically insignificant between-group differences in VO_2max_ and postexercise changes in this parameter in each subgroup, although a slight post-training rise of VO_2max_ (by 11.8%) was found in the experimental group.

A positive correlation coefficient was found between pre–post changes in the variable. No significant relations between ΔVO_2max_ and ΔT/C × 10^2^ were displayed in the control group (Table 7). It has turned out that the postexercise decrease in cortisol and the increase in the anabolic–catabolic index promotes the improvement of physical capacity.

Table 8 shows the changes in physical abilities in terms of the range of motion, strength, and endurance of selected muscle parts from two testing days in both groups. In the experimental group, the 8-week training resulted in significant improvements in the selected exercise capacity of the muscles of the upper body (abdominals), upper limbs (pull up), and lower limb power (SLJ, standing long jump). Interestingly, the control group also showed gains in the strength of the upper body and upper limb muscle and endurance, but these changes were much less significant.

## 4. Discussion

The dynamics of changes in the biosynthesis of endogenous testosterone depend on many factors. Among middle-aged men, higher testosterone levels have been observed in men reporting the highest level of physical activity [4,34,35]. The survey of the European Prospective Investigation into Cancer and Nutrition (EPIC), involving 696 men aged over 20 years, revealed significantly higher testosterone levels in men who claimed to be physically active for three or more hours per week compared to those who did not show greater physical activity [36]. The researchers demonstrated that there is an obvious link between physical activity and testosterone levels by examining more physically active men. The health benefits of testosterone result from the fact that it has a beneficial effect on the cardiovascular system by reducing the production of proinflammatory factors while increasing vascular endothelial regeneration [37,38]. Research on the effect of training type on the short- or long-term androgenic status is, therefore, important in health prevention in the middle-aged male population. Strength training has been shown to increase testosterone [39,40], whereas endurance training decreases its levels [41].

Biochemical blood tests confirmed the presented results as they showed significant changes in the experimental group (participants following the author’s training program) and no significant changes in the control group. It was demonstrated that regular participation in the research experiment significantly increased the values recorded during testosterone measurements. Since the strength and endurance training used in the present study had a positive effect on testosterone levels in a group of men aged 35–40 years, improvements in their health in terms of cardiovascular disease prevention can be anticipated. The strength and endurance training performed during the experiment resulted in an increase in testosterone levels, which, on the one hand, confirms the findings of the research on the relationship between strength training and testosterone levels. On the other hand, the novelty of our study is the increase in testosterone levels when strength training is combined with endurance training. It can be assumed that the achieved results, i.e., improvement of power, explosive strength, and aerobic capacity and the shift of metabolic balance towards anabolism, may have been influenced by the combination and sequence of exercises in each training session.

A more detailed study on strength training (resistance training) showed that in older men, the mean T:C ratio, 24 h after a variable intensity (VI) training session, was more than 33% higher than the value after a constant intensity (CI) session and that lower cortisol concentration after VI is responsible for this difference [42]. Similar tendencies were observed in young men [43]. This trend is in line with the observation of competitive kayakers who trained 3 times a week over a period of 3 weeks, in which 3 weeks of endurance training resulted in a slight decrease in T/C, whereas HIIT led to an increase in this parameter [44].

Some researchers suggest that at the beginning of the training period, hormonal responses should be examined immediately after a single training session. If the adrenal cortex responds excessively to training, it is suggested that the intervals be extended between repeated exercises, which allows a reduction of the stress response, i.e., glucocorticoid status, and, at the same time, achieves the planned physical load. In addition to the effect of the training period on resting hormonal status, it is important to determine the hormonal responses to a single training session consisting of repeated efforts. The results of hormonal tests performed before and immediately after the session will provide information about the magnitude of response to exercise stress and allow the modulation of the intensity and intervals between subsequent efforts to minimize the cortisol response. T/C measurements following a session with continuous aerobic exercise (CE) and after a session with intermittent exercise (IC) showed twice the T/C after the IC session [45]. Observations were made on T/C behavior during the three-hour postexercise recovery period. T/C fluctuated significantly during this period after a high-volume training session, whereas after a HIIT session, the T/C parameter was more stable. Taking into account the graph showing the area under the curve, it can be concluded that the time-integrated T/C value is greater after HIIT [46]. The only contradiction to the data presented above was reported by researchers when both continuous aerobic exercise and HIIT training sessions led to significant physical fatigue; in this case, 12 h after the end of the session, a greater decrease in T/C was noted after HIIT [47]. The results of our study on hormonal status, T, C, and T/C demonstrate that the proposed training program, continued for 8 weeks in a way that promotes protein metabolism, modulates the hormonal indices we are studying and slightly improves the parameter of physical capacity.

Our study confirmed all findings presented in the literature about the relationship between training and testosterone levels, which, for the experimental group, showed a substantial (statistically significant) increase after 8 weeks of training. In the control group, no statistically significant differences were found despite the increase in this parameter. It can also be assumed that this training program does not only lead to an increase in the level of this hormone but also improves the personal and sexual satisfaction, health, and physical fitness of the participants [26,48,49].

The results showed no post-training changes in aerobic capacity (VO_2max_), whereas significant improvements were observed in the muscle strength of the upper body, upper limbs, and lower limbs. This phenomenon is consistent with the current state of knowledge regarding training and its physiological and biomechanical responses. Workouts that are typically aerobic increase maximal oxygen uptake. HIIT sessions of sufficiently high intensity and frequency improve both aerobic and anaerobic capacity and fitness levels [25,50,51,52,53]. Typical resistance training improves only neural adaptation, which manifests itself as an increase in strength but does not alter VO_2max_. In light of the changes found, it can be concluded that the experimental group responded as if resistance exercises had a dominant role in the HIIT program.

The strength of our study design is the inclusion to the experiment of a group of similar age and anthropometric features who performed only their habitual physical activity over the same period. The results inspire further research on the optimization of HIIT and its monitoring based on physiological indices.

### Limitation of the Study

In the present study, we focused on evaluating the effects of high-intensity training on testosterone and cortisol levels. We did not investigate other physiological and biochemical mechanisms that may have been altered by the applied training protocol. The results of the study indicate that further research is needed to determine other body responses resulting from the proposed form of physical activity. Furthermore, it would be noteworthy to see how long, after the completion of the experiment, the favorable physical-hormonal status would persist if the experimental group returned to its former habitual activity.

## 5. Conclusions

The strength and endurance training, performed based on high-intensity interval sessions (circuit training), increases testosterone levels in men aged 35–40 years and can be used to enhance general well-being and partly inhibit harmful age-related changes.It is worth using this type of training in adult men because it can positively affect their quality of life and health by physiologically increasing testosterone levels, lowering cortisol, and improving anabolic–catabolic balance and muscle strength.This type of physical activity can act as an alternative or support pharmacotherapy for increasing testosterone levels in men.

## Figures and Tables

**Figure 1 jcm-10-02143-f001:**
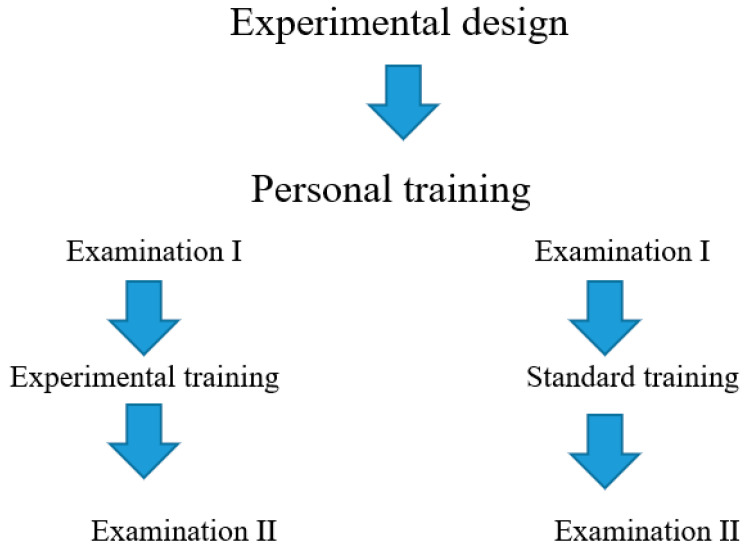
Training scheme. Source: own study

**Table 1 jcm-10-02143-t001:** Characterization of the study participants.

Group	*N*	Variable	Mean	SD	Min–Max	95%Cl
Experimental		Age (years)	37.7	1.9	35–40	36.81–38.78
	15	Stature (cm)	180.0	5.6	174–195	177.68–183.92
		BM (kg)	90.6	13.1	65.2–112.0	83.36–97.94
		BMI (kg/m^2^)	27.6	3.9	21.2–36.3	25.40–29.71
Control		Age (years)	37.8	1.7	35–40	36.88–38.17
	15	Stature (cm)	175.0	5.6	163–186	172.29–178.51
		BM (kg)	84.8	14.2	65.4–121.2	76.96–92.63
		BMI (kg/m^2^)	27.6	3.3	22.9–34.9	25.74–29.44

SD—standard deviation, CI—confidence interval, BM—body mass, BMI—body mass index.

**Table 2 jcm-10-02143-t002:** Description of exercises of the strength and endurance training program performed by the experimental group (Ex).

Interval Training 2:1 (HIIT)	Strength Circuit with the Example of Resistance Training with a Kettlebell
At the beginning of the training, participants had to do warm-up and adaptation exercises with minimal external resistance under the supervision of a personal trainer (duration 10–15 min). The participant performed a form of activity that involved high-intensity exercise alternated with rest (60 s/30 s). It consisted of 10 exercises that were performed one after the other to form a circuit. 1. Push-ups on dumbbells with the dumbbell pulled alternately to the chest at the moment of straightening the arms. 2. Squat with a dumbbell held with both hands at chest height. 3. Standing kettlebell side bends. 4. Jumps with a change of legs (from the forward lunge position). 5. Standing dumbbell press. 6. Half squat with a dumbbell held between legs with both hands. 7. Dumbbell weighted sit-ups. 8. Dumbbell pull (dumbbell row). 9. Push-ups. 10. Dumbbell reverse lunges.	The participant performed a small circuit of 5 exercises with a kettlebell (25 repetitions each). During one training session, they performed from 3 to 5 circuits, according to the principle of a gradual increase in loads. The rest between circuits was 1 to 2 min. 1. Kettlebell swing. 2. Standing one-handed press. 3. Squat with a kettlebell held with both hands in front of the body. 4. Kettlebell clean. 5. Kettlebell snatch. The duration of a training session was up to 30 min. The first session was devoted to learning and mastering the correct technique of the exercises. The participant ended each training session with stretching for about 6 min.

HIIT—high-intensity interval training, 2:1—two units of work to one unit of rest.

**Table 3 jcm-10-02143-t003:** Changes in morning serum testosterone levels (T) in the experimental and control groups following the 8-week training program.

Testosterone nmol/L	Experimental Group					Control Group					Between Groups		
	Mean	Median	Min	Max	SD	Mean	Median	Min	Max	SD	t_1_	*p*	Cohen’s d
Pre	14.85	14.23	8.05	24.05	4.09	16.66	16.76	8.02	27.62	5.10	−1.09	0.285	0.391
Post	20.30	18.60	13.85	25.85	4.23	17.66	18.46	8.05	26.82	5.45	1.48	0.150	0.541
differences	5.45	5.48	5.79	1.79	2.40	1.00	0.69	0.03	−0.80	2.32	**5.05**	**0.001**	1.89
Between examinations	**t_2_ = −9.08, *p* < 0.001,** Cohen’s d **=** 2.27					t_2_ = −1.53, *p* = 0.149, Cohen’s d = 0.435							

t_1_—Student’s *t*-test for independent samples; t_2_—Student’s *t*-test for dependent samples; *p*—likelihood ratio; Statistically significant values are in bold.

**Table 4 jcm-10-02143-t004:** Changes in morning serum cortisol levels (C) in the experimental and control groups following the 8-week training program.

Cortisol nmol/L	Experimental Group					Control Group					Between Groups		
	Mean	Median	Min	Max	SD	Mean	Median	Min	Max	SD	t_1_	*p*	Cohen’s d
Pre	460	428	334	666	93	390	406	201	593	113	**2.07**	**0.045**	0.681
Post	405	424	274	527	85	385	378	166	702	159	0.47	0.640	0.171
differences	−55	−4	−60	−139	118	−5	−28	−35	109	137	−1.21	0.235	0.391
Between examinations	t_2_ = −2.00, *p* < 0.061, Cohen’s d = 0.466					t_2_ = −0.13, *p* = 0.902, Cohen’s d = 0.037							

t_1_—Student’s *t*-test for independent samples; t_2_—Student’s *t*-test for dependent samples; *p*—likelihood ratio; significant values are in bold.

**Table 5 jcm-10-02143-t005:** Changes in the T-to-C ratio × 100 in the experimental and control groups following the 8-week training program.

T/C × 100	Experimental Group					Control Group					Between Groups		
	Mean	Median	Min	Max	SD	Mean	Median	Min	Max	SD	t_1_	*p*	Cohen’s d
Pre	3.17	2.97	2.16	4.18	0.79	4.63	4.12	2.39	8.33	1.71	**−3.37**	**0.002**	1.09
Post	5.05	4.91	3.55	7.61	1.13	5.39	5.00	1.77	13.18	2.73	−0.49	0.626	1.64
differences	1.88	1.94	1.39	3.43	1.19	0.76	0.88	−0.62	4.85	1.98	**2.11**	**0.042**	0.686
Between examinations	**t_2_ = −6.90, *p* ≤ 0.001,** Cohen’s d **= 1.58**					t_2_ = −0.198, *p* = 0.111, Cohen’s d = 0.384							

t_1_—Student’s *t*-test for independent samples; t_2_—Student’s *t*-test for dependent samples; *p*—likelihood ratio; significant values are in bold.

**Table 6 jcm-10-02143-t006:** Changes in the VO_2_ values in the exercising and nonexercising groups following an 8-week period.

VO_2max_ (mL/kg/min)	Experimental Group					Control Group					BetweenGroups		
	Mean	Median	Min	Max	SD	Mean	Median	Min	Max	SD	t_1_	*p*	Cohen’s d
Pre	32.2	32.0	27.4	38.1	3.6	32.8	22.7	26.3	41.2	4.3	1.08	0.288	**0.151**
Post	36.0	36.5	31.0	40.2	3.2	33.0	33.0	26.8	41.8	3.6	1.43	0.160	**0.881**
differences	3.8	3.5	0.5	10.9	−0.4	0.2	10.3	0.5	0.6	−0.7	0.17	0.869	**0.315**
Between examinations	t_2_ = 0.872, *p* = 0.395, Cohen’s d = **9.50**					t_2_ = −0.13, *p* = 0.902, Cohen’s d = 0.286							

t_1_—Student’s *t*-test for independent samples; t_2_—Student’s *t*-test for dependent samples; *p*—likelihood ratio; significant values are in bold.

**Table 7 jcm-10-02143-t007:** Linear regression describing the relationships between pre–post changes (Δ) in VO_2max_ T/C × 10^2^ values in the experimental group.

Regression Equation	Correlation (r)	*p*-Value
ΔVO_2max_ = 0.887 + 1.425 × ΔT/C × 10^2^	0.047	0.0047

**Table 8 jcm-10-02143-t008:** Changes in the selected parameters of physical fitness in the experimental (Ex) and control (Co) groups following an 8-week period.

Groups	Testing	Sit-Ups n/30 s	Pull-Ups	Standing Long Jump cm
Experimental Group	pre	20.6 ± 4.9	3.7 ± 4.2	208 ± 28
	post	26.4 ± 3.1	7.1 ± 4.3	222 ± 25
	difference	**t = −7.60, *p* = ≤0.001** **Cohen’s d = 1.93**	**t = −8.13, *p* = ≤0.001** **Cohen’s d = 2.09**	**t = −6.54, *p* = ≤0.001** **Cohen’s d = 1.69**
Control Group	pre	20.4 ± 4.8	2.7 ± 3.7	201 ± 16
	post	21.3 ± 4.7	3.1 ± 3.9	205 ± 18
	difference	**t = −2.22, *p* = 0.043** **Cohen’s d = 0.574**	**t = −2.44, *p* = 0.028** **Cohen’s d = 0.634**	t = −1.38, *p* = 0.195 Cohen’s d = 0.355

t_1_—Student’s *t*-test for independent samples; t_2_—Student’s *t*-test for dependent samples; *p*—likelihood ratio; significant values are in bold.

## Data Availability

The data presented in this study are available on request from the corresponding author.

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
