# Peer review of "The Effect of High-Intensity Interval Training Periods on Morning Serum Testosterone and Cortisol Levels and Physical Fitness in Men Aged 35–40 Years"

_jcm, 2021, doi:10.3390/jcm10102143_

Round 1
Reviewer 1 Report
Overall comments:
The introduction of this manuscript was written to a very good standard. This section leads the reader clearly towards the research aim of this study. There is opportunity to include some information about the type of physical activity intervention being competed in this study. The reader needs to know the impact that this physical activity may have on the hormonal pathways the author(s) are examining in the research study.
The methods section is completed to a good standard. There could be a clearer indication of the numbers in each group and how the participants were choosen for each group.
Results: It is more common to see the presentation of statistical analysis in the Results text which is supported by the tables (or figures). Here the text just directs the reader towards the tables and outlines what information the tables are providing. This should be clear from the title of the table itself. The correlation figures appear as if taken directly from a statistical package. This would not be normal practice.
Specific comments
Introduction
Page 2
Line 52: This is a little off from the topic of the research article. Yes, you are examining testosterone in response to an 8-week training program but here you refer to doping with exogenous testosterone for muscle and strength gain. In my opinion, this distracts the reader away from your topic a little.
Line 74: The health effects of hypogonadism highlighted here are experienced in older populations. This does not align with the 35 – 40 year bracket you are examining. Perhaps it is best to highlight the impact of hypogonadism is with this older adult population.
Line 91: More information on the role of cortisol in the chronic health conditions noted in the manuscript could be useful here. Consider why excessive cortisol can lead to some of the chronic health conditions referred to in the manuscript. Consider the relationship between excessive cortisol and its accepting receptors. It is known that cortisol (glucocorticoids) can affect the vasculature by attaching to mineralocorticoid receptors if cortisol is in excess leading to hypertension.
Page 3
Line 113
This would seem like an appropriate area to add in a short section on high intensity intermittent exercise. For example, would an acute exercise of such an exercise mode lead to an activation of the HPA and HPG axes to induce a change in circulating cortisol and testosterone? If so, would repeated exposure possible lead to an improved (elevated)
What about cortisol and health ageing? It is known that it can elevate with healthy ageing. This would support the health aspect of this paper.
Materials and Methods
Table 1: It is not clear how many participants are in each group (Experimental and Control). It is noted that the study involved 30 participant. The assumption is that each group contains 15 participants.
Table 1: Alter “Age (ys)” to “Age (years)” or “Age (y)”.
Line 139: It is noted that there are 15 participants in each group. This still should be highlighted in Table 1 as per the earlier comment.
Line 159: Fix the jump in text here.
Line 171 The abbreviation VO2max has already been defined. Use the abbreviation from here onwards.
Line 161 & Line 204: Is there any way to confirm that the 8 week training program was more intense for the experimental group? Did they also complete their normal physical activity alongside this training program. Was there an example week of training examined before the study and compared to a week of training within the program?
When the measurement of “lifestyle” is detailed in Line 161 does this mean physical activity? If so, how was this measured?
Line 205: Alter “in” to “of”.
Results
It is more common to see the presentation of statistical analysis in the Results text which is supported by the tables (or figures). Here the text just directs the reader towards the tables and outlines what information the tables are providing. This should be clear from the title of the table itself. The correlation figures appear as if taken directly from a statistical package. This would not be normal practice.
Table 3 & 4: What does “O41” refer to ?
Table 6: Ensure all numerical values are presented with a “.” and not a “,”.
Figure 1: Same comment as above regarding the use of a decimal place.
Figure 1 and Figure 2: These could be presented in a clearer and cleaner fashion. These look like they have been taken directly from the statistical package.
Page 6
Line 2 of section: alter 2.5kmhr-1, also add spaces between numbers and units throughout.
Table 2 and 3 Recheck the title of the tables. It refers to percentage increases but the HR measure is not a percentage increase it is an absolute change.
Page 8
Line 248: The abbreviation “Ex” should be presented in the methodology?
Discussion
Page 9
Line 283: There should be a summary to the findings in the first paragraph of the discussion.
Page 10
Line 320: “wheffortsich”?
Line 321: Add “In”.
There is a need for a consideration why exercise has lead to the increase in testosterone and a reduction in cortisol. It is not clearly laid out in this discussion. The structure needs refining and content added to support the findings.
Line 379: Taking exogenous testosterone will lead to greater elevations in testosterone that is seen in your own study. I do not think you study supports the point that this could be an alternative to doping. Doping in this manner is completed for many different reasons and do not align with the moderate changes seen in your study. I do, however, believe you study shows a 8-week training (HIIT) programme can improve testosterone responses and lower cortisol activity in young (35 – 40 years old). This may be impactful on future possible declines in testosterone and elevations of cortisol that are seen with healthy ageing but are links to some chronic health conditions. Personally, I think you should not include doping into this manuscript as I think it detracts from some of the messages of your own findings.
Author Response
Dear Reviewer,
Thank you very much for your time and valuable comments, which all have been considered and incorporated. The detailed list of responses is given below. We hope that the modifications and explanation will be acceptable for you.
Yours sincerely,
Rydzik, corresponding author
Overall comments:
The introduction of this manuscript was written to a very good standard. This section leads the reader clearly towards the research aim of this study. There is opportunity to include some information about the type of physical activity intervention being competed in this study. The reader needs to know the impact that this physical activity may have on the hormonal pathways the author(s) are examining in the research study.
A: Thank you very much for this comment, but to meet the requirements of the other Reviewers, we have modified this section, which resulted in a shorter version. However, information on physical activity and its relationships with physiological and hormonal responses in middle-aged people has been added.
The methods section is completed to a good standard. There could be a clearer indication of the numbers in each group and how the participants were choosen for each group.
A:The size of both groups (N) has been added in Table 1. Group selection was randomized using a number generator. This information is contained in the text under the Methods section.
Results: It is more common to see the presentation of statistical analysis in the Results text which is supported by the tables (or figures). Here the text just directs the reader towards the tables and outlines what information the tables are providing. This should be clear from the title of the table itself. The correlation figures appear as if taken directly from a statistical package. This would not be normal practice.
The regression equations and values of r and p can be reported in the Results section.
Specific comments
Introduction
Page 2
Line 52: This is a little off from the topic of the research article. Yes, you are examining testosterone in response to an 8-week training program but here you refer to doping with exogenous testosterone for muscle and strength gain. In my opinion, this distracts the reader away from your topic a little.
A:The entire section of the text poorly related to the problems studied including line 52 has been removed
Line 74: The health effects of hypogonadism highlighted here are experienced in older populations. This does not align with the 35 – 40 year bracket you are examining. Perhaps it is best to highlight the impact of hypogonadism is with this older adult population.
A:Indeed, persistent hypogonadism occurs mainly in older men where decreased testosterone concentrations and increased SHBG concentrations of specifically binding molecules of sex hormone T and estradiol are observed. As a result, the concentration of the biologically active T fraction is much lower than it is when SHGB levels match the reference range. On the other hand, in the middle-aged population, there are cases of too low resting T (<7.0 mM) due to working through a long-term mental stress. When this condition lasts longer, SHBG levels rise and the androgenic status is similar to that of older men. Pseudo-hypogonadism (reduced T concentrations) is observed in endurance athletes ( Hackney AC , Szczepanowska E, Viru A 2003) and in late season in wrestlers who follow a restrictive diet during the training and competition season. In our study, a total of 4 cases of very low T close to the critical value (8.1, 8.0, 8.9, 8.1 nM), were reported in all examinations and groups. This part has been modified
Line 91: More information on the role of cortisol in the chronic health conditions noted in the manuscript could be useful here. Consider why excessive cortisol can lead to some of the chronic health conditions referred to in the manuscript. Consider the relationship between excessive cortisol and its accepting receptors. It is known that cortisol (glucocorticoids) can affect the vasculature by attaching to mineralocorticoid receptors if cortisol is in excess leading to hypertension.
A:Hypercortisolism was not emphasized much in the paper. It occurs in disease states (e.g., Cushing syndrome, adrenal tumor, hyperpituitarism). In our study, there were a total of 3 cases of C concentrations slightly above the reference range (666 and 2 x 702 nmol/L). It is assumed that concentrations exceeding 550 nmol/L at normal concentrations of corticosteroid-binding globulin (CBG) are responsible for slightly elevated free C and therefore higher glucocorticoid activity. The same occurs when a CBG variant of the lowered binding affinity is present in the blood. This part has been modified
Page 3
Line 113:
This would seem like an appropriate area to add in a short section on high intensity intermittent exercise. For example, would an acute exercise of such an exercise mode lead to an activation of the HPA and HPG axes to induce a change in circulating cortisol and testosterone? If so, would repeated exposure possible lead to an improved (elevated)
A:We have added relevant literature with comments regarding the effect of training type on HPG but not HPA. Hyperactivity of HPA during annual training season was noted only in females (rowers) who belonged to top-level competitive athletes (Vervoorn 1991/1992)
What about cortisol and health ageing? It is known that it can elevate with healthy ageing. This would support the health aspect of this paper.
A:It is very important issue . Prolonged exposure of human brain to higher cortisol may result in neurodegenerative changes in hippocamp and in consequences lost memory and other cognitive functions.
Materials and Methods
Table 1: It is not clear how many participants are in each group (Experimental and Control). It is noted that the study involved 30 participant. The assumption is that each group contains 15 participants.
A: Table 1 has been changed according to the Reviewer's suggestions, age has been modified, and (N) has been added
Table 1: Alter “Age (ys)” to “Age (years)” or “Age (y)”.
A: This part has been changed.
Line 139: It is noted that there are 15 participants in each group. This still should be highlighted in Table 1 as per the earlier comment
A: This has been corrected
Line 159: Fix the jump in text here.
A: This has been corrected
Line 171 The abbreviation VO2max has already been defined. Use the abbreviation from here onwards.
A: This has been corrected
Line 161 & Line 204: Is there any way to confirm that the 8 week training program was more intense for the experimental group? Did they also complete their normal physical activity alongside this training program. Was there an example week of training examined before the study and compared to a week of training within the program?
A: The 8-week training program was conducted by a professional personal trainer and exercise intensity in the experimental group was determined by measuring heart rate. The control group continued their previous recreational physical activity, and their intensity was assessed similarly. Furthermore, a personal trainer who conducted the experiment monitored the activity level of both groups and evaluated the training intensity by means of subjective perception of the effort level of the participants. Based on the interviews, measurements, and observations, a significant increase in exercise intensity was found in the experimental group. HIT training itself is a very intense form of training[1,2].
- Menz, V.; Marterer, N.; Amin, S.B.; Faulhaber, M.; Hansen, A.B.; Lawley, J.S. Functional Vs. Running Low-Volume High-Intensity Interval Training: Effects on VO2max and Muscular Endurance. J. Sports Sci. Med. 2019, 18, 497–504.
- Paul Laursen; Martin Buchheit Science and Application of High-Intensity Interval Training; Human Kinetics Publishers, 2019; ISBN 978-1-4925-5212-3.
When the measurement of “lifestyle” is detailed in Line 161 does this mean physical activity? If so, how was this measured?
A:This means lifestyle by verifying the diets, sleep duration, supplementation, and stimulants.The participants provided the relevant information by recording and during interviews. Physical activity was only performed in the experiment
Line 205: Alter “in” to “of”.
A: This has been corrected
Results
It is more common to see the presentation of statistical analysis in the Results text which is supported by the tables (or figures). Here the text just directs the reader towards the tables and outlines what information the tables are providing. This should be clear from the title of the table itself. The correlation figures appear as if taken directly from a statistical package. This would not be normal practice.
A: Figures have been removed and replaced with a table. Descriptions have been modified
Table 3 & 4: What does “O41” refer to ?
A: This has been deleted
Table 6: Ensure all numerical values are presented with a "." and not a ",".
A: This has been corrected
Figure 1: Same comment as above regarding the use of a decimal place.
Figure 1 and Figure 2: These could be presented in a clearer and cleaner fashion. These look like they have been taken directly from the statistical package.
A: The figures have been removed and replaced by Table 7
Page 6
Line 2 of section: alter 2.5kmhr-1, also add spaces between numbers and units throughout.
A: This has been corrected
Table 2 and 3 Recheck the title of the tables. It refers to percentage increases but the HR measure is not a percentage increase it is an absolute change.
A: Table titles have been checked and verified
Page 8
Line 248: The abbreviation “Ex” should be presented in the methodology?
A: This has been corrected
Discussion
Page 9
Line 283: There should be a summary to the findings in the first paragraph of the discussion.
A: This has been complemented
Page 10
Line 320: “wheffortsich”?
A: This typo has been corrected
Line 321: Add “In”.
A: This word has been added
There is a need for a consideration why exercise has lead to the increase in testosterone and a reduction in cortisol. It is not clearly laid out in this discussion. The structure needs refining and content added to support the findings.
Probably moderate program of exercises has beneficial effect on mood state. There are many proofs that directly following not exhaustive exercises state anxiety tended to decrease. That explain the syndrome of addiction to exercise. Unfortunately we did not conduct parallel psychological study. The other explanation of post-training decline of cortisol is adaptation of HPA axis to any stimulus (down regulation)
A: The discussion has been supplemented
Line 379: Taking exogenous testosterone will lead to greater elevations in testosterone that is seen in your own study. I do not think you study supports the point that this could be an alternative to doping. Doping in this manner is completed for many different reasons and do not align with the moderate changes seen in your study. I do, however, believe you study shows a 8-week training (HIIT) programme can improve testosterone responses and lower cortisol activity in young (35 – 40 years old). This may be impactful on future possible declines in testosterone and elevations of cortisol that are seen with healthy ageing but are links to some chronic health conditions. Personally, I think you should not include doping into this manuscript as I think it detracts from some of the messages of your own findings.
A:We completely agree, this topic has been removed from the text.
Reviewer 2 Report
Dear author,
I would like to congratulate you on the effort to carry out this study, in a clear attempt to bridge the gap between literature and application to decision making among exercise physiologists, strength and conditioning coaches or personal trainers. In my opinion the overall appreciation of the manuscript is highly positive, as the present work can provide an interesting, up to date and evidence-based model, particularly directed to a non-sporting population, such as elderly adults. I find the manuscript displayed in a generally well-written English, with an acceptable flow throughout the text, although it could be improved. Nevertheless, I have some concerns or comments for your consideration, particularly regarding the consistency in some raised aspects of the rationale, and the potential benefits and limitations, as following:
Abstract
Lines 18–22: The background is too long and not objective. I strongly suggest an improvement.
Lines 22–29: Although limited by the number of allowed characters, further information is needed in order to briefly demonstrate what was done and how was done.
Lines 29–32: When relevant, I suggest adding some values, in order to highlight the relevance of the results.
Line 35: I suggest the author to change the keywords, since most of them are already included in the title, thus providing less chances to find the study in a future search on databases.
- Introduction
The introduction section is not clear when supporting the potential reader with a proper framework for the present study, providing several elements quite confusing for its rationale, that are not highlighted in the following sections.
Lines 38–76: Please be more concise and objective when providing the framework supporting the relevance of the present study, or the selected age range of the sample. Lines 52 to 53 clearly demonstrate and example of information that is not relevant in the present context. Please consider a structural change throughout this section and improving the text flow. Another argument of upmost importance when stating the somewhat irrelevance of some aspects of the introduction is that most of the used citations are not exploited in the discussion section.
Lines 77–118: I believe that this is the core framework for the present work. This should be explored by including information of what is known about high-intensity training and endocrine system, and why it remains important to clear the effects of a concurrent type of training program when compared to a circuit strength training.
- Material and Methods
2.1. Study group
Could the authors provide a clearer overview regarding the pilot study: what was done and how it influenced the decisions on the following procedures.
Table 1: Please correct kg/m2 to kg/m2 (superscript). Also, please add 95% confidence intervals, whenever possible.
Lines 133–135: I find this paragraph quite confusing. Please change.
Lines 137–139: How was the randomization procedures performed? Were participants and observers blinded regarding the opposite group training and expectations? Further information and clarity are needed. Please include.
2.2. Research program and methodology
If possible, I suggest including a graphical element, representing the study timeline or roadmap, and testing moments and variables, in order to provide the potential reader with a clearer suggestion about the procedures.
Line 143: Could you inform about what the previous program of recreational physical activity consisted about?
Line 158: ICC, %CV and relationship between post exercise changes and cortisol levels and anabolic-catabolic balance index and Vo2max was performed. Please include here.
Line 159: It is strongly suggested that authors include power analysis.
Measurement of physical fitness
Line 178: Please correct to km∙h-1.
Line 179: Please correct to km∙h-1.
Line 183: Could the authors inform about other criteria considered for the VO2max test?
2.3. Experimental program
Line 205: Confusing. Please change this sentence.
Lines 206–211: actually, the type of training used was a concurrent training, adopting a high-intensity programming. What features makes this HIIT program so unique and differentiated? Also, please provide a clear distinction between both training programs. Further explanations are needed.
- Results
Line 226: I believe you meant Figure 1. Please correct.
Line 228: please consider the following change - “…and VO2max among the experimental group participants.”
Table 3: I find it quite odd the difference in the morning serum testosterone levels between the experimental group and the control group. The lower mean and median values could bias the observed results in the post testing differences, particularly in the T-to-C ratio. Please note and discuss this observation as it can seriously affect the interpretations and implications of the study.
Tables 3, 4 and 5: These elements should be reorganized and clearer. What does it mean p<0.061 or p<0.395? Please insert the correct value, if possible. Please change p<0.000 to p≤0.001. Also, include the effect size, whenever possible.
Table 7: Please change p<0.000 to p≤0.001.
- Discussion
I have serious concerns regarding the general discussion and interpretation of the observed results as it could be biased as a result of the randomization process and initial values of testosterone.
Line 229: Again, I have some concerns with this observation, because differences were already installed in the pre test and this fact could explain the noted differences in the post test for the testosterone levels and T-to-C index.
Line 315: “(endurance tr)” – did you meant training? Please correct.
Line 320: “wheffortsich” – what does this mean? Please correct.
Lines 344–346: Not all these variables were assessed, so one can only speculate, assuming the effects of high testosterone levels. Cautions should be taken.
Line 347: This is not consistent with what was reported in Line 31. Please correct.
Line 350: Please delete the extra “.”.
- Conclusions
Lines 376–378: Probably, but the present study results do not allow one to infer such a thing.
References
It seems that less references would be more appropriated in order to provide an objective and precise approach throughout the manuscript. Most of the time, authors are not clear and straight to point.
Author Response
Dear Reviewer,
Thank you very much for your time and valuable comments, which all have been considered and incorporated. The detailed list of responses is given below. We hope that the modifications and explanation will be acceptable for you.
Yours sincerely,
Rydzik, corresponding author
Abstract
Lines 18–22: The background is too long and not objective. I strongly suggest an improvement.
A: Thanks for the comments. These lines have been removed
Lines 22-29: Although limited by the number of allowed characters, further information is needed in order to briefly demonstrate what was done and how was done.
That is just added
Lines 29-32: When relevant, I suggest adding some values, in order to highlight the relevance of the results.
Line 35: I suggest the author to change the keywords, since most of them are already included in the title, thus providing less chances to find the study in a future search on databases.
The entire abstract has been corrected according to the suggestions
- Introduction
The introduction section is not clear when supporting the potential reader with a proper framework for the present study, providing several elements quite confusing for its rationale, that are not highlighted in the following sections.
A: This has been corrected
Lines 38-76: Please be more concise and objective when providing the framework supporting the relevance of the present study, or the selected age range of the sample. Lines 52 to 53 clearly demonstrate and example of information that is not relevant in the present context. Please consider a structural change
throughout this section and improving the text flow. Another argument of upmost importance when stating the somewhat irrelevance of some aspects of the introduction is that most of the used citations are not exploited in the discussion section.
A:The Introduction has been corrected. The number of citations and related text volume has been reduced to achieve the appropriate balance between Introduction and discussion.
Lines 77-118: I believe that this is the core framework for the present work. This should be explored by including information of what is known about high-intensity training and endocrine system, and why it remains important to clear the effects of a concurrent type of training program when compared to a circuit strength training.
A: Thank you, the introduction has been thoroughly rewritten according to the suggestions from all reviewers. In our opinion, the introduction is now more concise, objective, and presents a framework to support the relevance of the research and the selected age range, and the associations of high-intensity training with the response of the hormonal system.
- Material and Methods
2.1. Study group
Could the authors provide a clearer overview regarding the pilot study: what was done and how it influenced the decisions on the following procedures.
A: In the pilot study, a group of men was qualified for the experiment based on basic physical fitness examinations using the European Fitness Test (Eurofit) (Szopa et al. 1996). This qualification aimed to maintain the principles of purposive selection, which assumed that the subjects had an initial level of strength and conditioning (they trained recreationally). The initial strength and conditioning base of the participant was necessary because the experimental training program (HIIT) was characterized by high intensity. Furthermore, both the qualification and supervision of exercisers were handled by a professional personal trainer.
Table 1: Please correct kg/m2 to kg/m2 (superscript). Also, please add 95% confidence intervals, whenever possible.
A: Confidence intervals have been added and superscripts have been corrected
Lines 133-135: I find this paragraph quite confusing. Please change.
A: The text has been modified
Lines 137-139: How was the randomization procedures performed? Were participants and observers blinded regarding the opposite group training and expectations? Further information and clarity are needed. Please include.
A: Group selection was randomized using a number generator. This information is contained in the text under the Methods section.
Randomization process: The purposive pre-selected study group was randomly assigned to two groups, with one group (Ex) subjected to the evaluated new procedure and the other group being the comparison group. Groups were not informed of the researchers' expectations. Their task was to strictly follow their training regimen and monitor their lifestyles. The parallel-group technique was used.
2.2. Research program and methodology
If possible, I suggest including a graphical element, representing the study timeline or roadmap, and testing moments and variables, in order to provide the potential reader with a clearer suggestion about the procedures.
Line 143: Could you inform about what the previous program of recreational physical activity consisted about?
A:Up to the beginning of the experiment, all subjects were engaged in recreational physical activity in the form of training (running, cycling, swimming, gym). The important information for us was that the activity so far was regular (at least twice a week).
Line 158: ICC, %CV and relationship between post-exercise changes and cortisol levels and anabolic-catabolic balance index and Vo2max was performed. Please include here.
A: Relevant information has been added
Line 159: It is strongly suggested that authors include power analysis.
A: This has been added in Table 7
Measurement of physical fitness
Line 178: Please correct to km∙h-1.
Line 179: Please correct to km∙h-1.
A: This has been corrected
Line 183: Could the authors inform about other criteria considered for the VO2max test?
A: The main criterion for the assessment of VO2max in the test was the analysis of the kinematics of respiratory changes. Other cardiorespiratory indices were analyzed during the test, which were not used by the authors of the present study. Maximum oxygen uptake is considered by the authors only an indicator of the effect of post-exercise changes in aerobic capacity.
2.3. Experimental program
Line 205: Confusing. Please change this sentence.
A: This has been corrected
Lines 206–211: actually, the type of training used was a concurrent training, adopting a high-intensity programming. What features makes this HIIT program so unique and differentiated? Also, please provide a clear distinction between both training programs. Further explanations are needed.
A: This part has been corrected by adding more information about control group activity. The explanation of the uniqueness and variability of HIIT training has been presented in detail in the experiment description and Table 2.
- Results
Line 226: I believe you meant Figure 1. Please correct.
A: Due to the suggestions of other Reviewers, the figures have been replaced by Table 7
Line 228: please consider the following change - "...and VO2max among the experimental group participants."
A: Due to the suggestions of other Reviewers, the figures have been replaced by Table 7
Table 3: I find it quite odd the difference in the morning serum testosterone levels between the experimental group and the control group. The lower mean and median values could bias the observed results in the post testing differences, particularly in the T-to-C ratio. Please note and discuss this observation as it can seriously affect the interpretations and implications of the study.
A: We did not have an influence on the baseline testosterone levels and we randomized participants. However, it should be noted that the minimum values in both groups were similar, the maximum value differed, but this could have been caused by one person. However, the effect of the applied experimental training and the increase of testosterone in the study group, which is many times greater than in the control group (difference 5.45 vs 1.00) as confirmed by statistical significance at p<0.001 seems to be the most noticeable. A similar situation occurs for the T/C ratio.
Tables 3, 4 and 5: These elements should be reorganized and clearer. What does it mean p<0.061 or p<0.395? Please insert the correct value, if possible. Please change p<0.000 to p≤0.001. Also, include the effect size, whenever possible.
A: This has been corrected. Effect size added
Table 7: Please change p<0.000 to p≤0.001.
A: This has been corrected
- Discussion
I have serious concerns regarding the general discussion and interpretation of the observed results as it could be biased as a result of the randomization process and initial values of testosterone.
Line 229: Again, I have some concerns with this observation, because differences were already installed in the pre test and this fact could explain the noted differences in the post test for the testosterone levels and T-to-C index.
A: We have attempted to improve the discussion based on the difference being the training effect.
Line 315: “(endurance tr)” – did you meant training? Please correct.
A: This has been corrected
Line 320: “wheffortsich” – what does this mean? Please correct.
A: This has been corrected
Lines 344-346: Not all these variables were assessed, so one can only speculate, assuming the effects of high testosterone levels. Cautions should be taken.
A: The conditional mood has been used to be cautious. We tried to support the speculation with literature
Line 347: This is not consistent with what was reported in Line 31. Please correct.
A: This has been corrected
Line 350: Please delete the extra “.”.
A: Removed
- Conclusions
Lines 376-378: Probably, but the present study results do not allow one to infer such a thing.
A: The conclusion is written in the conditional mood and based on conjecture and interpretation of the results. It also encourages further research in this area.
References
It seems that less references would be more appropriated in order to provide an objective and precise approach throughout the manuscript. Most of the time, authors are not clear and straight to point.
A: We have reduced and modified the literature
Round 2
Reviewer 2 Report
Dear authors,
I would like to congratulate you on the effort to carry out this new version of the study. It stands faithful to its clear attempt to bridge the gap between literature and application to decision making among exercise physiologists, strength and conditioning coaches or personal trainers. In my opinion the overall appreciation of the manuscript is highly positive, as the present work can provide an interesting, up to date and evidence-based model, particularly directed to a non-sporting population, such as elderly adults. I find the updated version of the manuscript suitable for the Journal of Clinical Medicine standards.
Abstract
The authors acknowledged all the addressed comments and suggestions and I find now the abstract appropriate and clear.
- Introduction
Authors made a meritorious effort to be more concise and objective when providing the framework supporting the relevance of the present study. In the updated version of the introduction section the selected age range of the sample was clearly presented as a need to examine. Structural changes have been performed throughout the section improving the text flow. Also, the number of citations and the references used are now suitable for the core framework for the present work.
- Material and Methods
2.1. Study group
After reding the authors response regarding the pilot study, I believe it become clearer for me. As a suggestion, maybe this information could be also important for the potential reader, although changes have been made in lines 289–293.
Table 1: Present 95% CI in a single column as (36.81 to 38.66) or (36.81–38.66)
2.2. Research program and methodology
I am satisfied with the performed changes and find It suitable.
Measurement of physical fitness
Again, authors’ acknowledged the reviewer suggestions and integrated it in this section.
2.3. Experimental program
Line 205: Confusing. Please change this sentence.
Lines 437–438: “The control group (Co) pursued their previous recreational physical activity that was monitored but not programmed by the authors of this study”. Please correct.
- Results
I found the performed changes appropriate and suitable for the study aimed and well conducted, providing the necessary elements do interpretate and discuss in the following sections.
- Discussion
The discussion has improved in the present version.
- Conclusions
The third conclusion was clearly necessary and is now in accordance with what was performed during the present study.
References
A clear improvement was made, and the overall quality of the manuscript benefited from it.
Author Response
Dear Reviewer,
Thank you very much for your time and valuable comments, which all have been considered and incorporated. The detailed list of responses is given below. We hope that the modifications and explanation will be acceptable for you.
Yours sincerely,
Rydzik, corresponding author
I would like to congratulate you on the effort to carry out this new version of the study. It stands faithful to its clear attempt to bridge the gap between literature and application to decision making among exercise physiologists, strength and conditioning coaches or personal trainers. In my opinion the overall appreciation of the manuscript is highly positive, as the present work can provide an interesting, up to date and evidence-based model, particularly directed to a non-sporting population, such as elderly adults. I find the updated version of the manuscript suitable for the Journal of Clinical Medicine standards.
A: Thank you very much.
Abstract
The authors acknowledged all the addressed comments and suggestions and I find now the abstract appropriate and clear.
A: Thank you
- Introduction
Authors made a meritorious effort to be more concise and objective when providing the framework supporting the relevance of the present study. In the updated version of the introduction section the selected age range of the sample was clearly presented as a need to examine. Structural changes have been performed throughout the section improving the text flow. Also, the number of citations and the references used are now suitable for the core framework for the present work.
A: Thank you
- Material and Methods
2.1. Study group
After reding the authors response regarding the pilot study, I believe it become clearer for me. As a suggestion, maybe this information could be also important for the potential reader, although changes have been made in lines 289–293.
A: Updated
Table 1: Present 95% CI in a single column as (36.81 to 38.66) or (36.81–38.66)
A: This has been corrected
2.2. Research program and methodology
I am satisfied with the performed changes and find It suitable.
A: Thank you
Measurement of physical fitness
Again, authors’ acknowledged the reviewer suggestions and integrated it in this section.
A: Thank you
2.3. Experimental program
Line 205: Confusing. Please change this sentence.
A: This has been corrected
Lines 437–438: “The control group (Co) pursued their previous recreational physical activity that was monitored but not programmed by the authors of this study”. Please correct.
A: This has been corrected
- Results
I found the performed changes appropriate and suitable for the study aimed and well conducted, providing the necessary elements do interpretate and discuss in the following sections.
A: Thank you
- Discussion
The discussion has improved in the present version.
Thank you
- Conclusions
The third conclusion was clearly necessary and is now in accordance with what was performed during the present study.
A: Thank you
References
A clear improvement was made, and the overall quality of the manuscript benefited from it
A: Thank you
This manuscript is a resubmission of an earlier submission. The following is a list of the peer review reports and author responses from that submission.
Round 1
Reviewer 1 Report
Abstract:
Line 22, Please identify whether the control group was inactive. The current expression reads as though both groups participated in HIIT.
Line 25-30: There is no explanation for how a "slight" change is defined. Is this based on the use of effect size or some other measure? It would also be useful to identify whether this decrease was significant.
Introduction: The introduction is well written. However, some consideration for restructuring this section to focus on the health risks and needs of the 30-40-year-old population group examined would be useful. The latter portions of the current paragraph best highlight why this study is important and could be signposted earlier.
Line34-43: This section focuses largely on older age and the effects of this. This does not seem relevant given that the population examined in this study is aged 30-40 years. Please amend the focus of this section to address the specific aging-related issues with this population.
Line 59: Please provide a reference to support the diagnosis of hypogonadism and the links to depressive states.
Line 65: Please provide a reference to support the use of exercise and lifestyle modification as an alternate therapy to testosterone supplementation.
Line 71: It would be useful to note some of the "health-promoting effects" that you refer to here. This will aid readers in understanding why you have chosen to evaluate cortisol, testosterone and Vo2max as outcomes.
Materials and Methods
Line 103: Please confirm that participants were all aged 35-40 years. If so, the title of the article seems misleading.
Line 116: It is unclear which portions or not, the control group participated in. Please make reference to a published protocol or more clearly describe the specific aspects of each group's engagements.
Line 121: The abbreviation T for testosterone has already been defined. The abbreviation C for cortisol should be identified at its first use in the introduction.
Line 143: Please identify a reference to support the statement "considered to be the most effective in the 21st century".
Line 152: It is unclear how you determined the pace of exercise and stage of training. Please provide clarity.
Table 1: Is there a reason that two variations of push-ups were included? Why not an incline bench press or similar to target alternate musculature?
There is no descriptor provided for measuring the intensity of this circuit.
Please identify in the table which components relate to the control group and which relate to the intervention group.
Results:
Where are the demographic details for each group? These should be provided given that the outcomes of interest are affected by age.
Table 3: Given that there is a difference between groups at baseline, it is important that you provide demographic data which may aid in understanding this.
You note in the abstract that there was a "slight" change in cortisol in the intervention group. Given that p = 0.061, we should consider that there is no difference (statistically). You may later interpret this reduction as being clinically meaningful with appropriate evidence.
Table 5: There are no differences in Vo2max. To suggest that a "slight" increase was evident in this measure in the experimental group is inappropriate. A difference in the mean Vo2 of 80 ml/min is negligible and likely within the error of measurement for the system.
Discussion
The first paragraph of the discussion does not seem to relate to any explanation of the results section. Please revise.
Please avoid the use of the term "insignificant". Rather, present as not significant. Whilst changes might not be statistically significant, that does not necessarily render them to be insignificant.
Line 228-230: Since the strength and endurance training used in the present study had a positive effect on testosterone levels in a group of men aged 35-40 years, improvements in their health in terms of cardiovascular disease prevention can be anticipated. Given that the majority of men in the study were already in the normal range for T, is it likely that they will experience any significant effects of this?
Line 236-242: This section speaks to older men (age range not defined) and does not appear to be relevant to the current investigation.
Line 243-248: Why is it important to modulate the T/C ratio based on an acute response? The issue with increased cortisol or decreased T comes from chronic exposure to undesirable measures of each. Further, It would be relevant to consider that T fluctuates across the day in line with circadian rhythms. Thus, the timing of any increase in T might be more crucial than the amount in order to address daily fluctuations and the androgenic response to the exercise stimulus.
The discussion section needs to address what appears to be marginal if not inconsequential changes in Vo2max. A mean difference of less than 1 metabolic equivalent suggests that the exercise intervention had no effect on fitness.
Some discussion of the baseline difference in C between groups should be included here. Could this be related to demographic factors? The authors should also consider whether the lower C levels led to a floored effect for change in this outcome within the control group. Reporting of normal maximal and minimal range for this measure would assist in presenting this and its explanation.
Given that the intervention programs were resistance training-based, it would be logical to include information on changes in strength. This may prove more useful in relation to the measurement of testosterone and cortisol and their anabolic/catabolic actions.
Author Response
Dear Reviewer,
Thank you very much for your time and valuable comments, which all have been considered and incorporated. The detailed list of responses is given below. We hope that the modifications and explanation will be acceptable for you.
Yours sincerely,
Rydzik, corresponding author
Line 22, Please identify whether the control group was inactive. The current expression reads as though both groups participated in HIIT.
A: This has been corrected
Line 25-30: There is no explanation for how a "slight" change is defined. Is this based on the use of effect size or some other measure? It would also be useful to identify whether this decrease was significant.
A: This has been corrected
Introduction: The introduction is well written. However, some consideration for restructuring this section to focus on the health risks and needs of the 30-40-year-old population group examined would be useful. The latter portions of the current paragraph best highlight why this study is important and could be signposted earlier.
A: This has been corrected and more references have been referred to.
Line34-43: This section focuses largely on older age and the effects of this. This does not seem relevant given that the population examined in this study is aged 30-40 years. Please amend the focus of this section to address the specific aging-related issues with this population.
A:Thank you for your comment, we have modified the text by complementing it with information related to middle-aged people.
Line 59: Please provide a reference to support the diagnosis of hypogonadism and the links to depressive states.
A: References have been added.
Line 65: Please provide a reference to support the use of exercise and lifestyle modification as an alternate therapy to testosterone supplementation.
A: References have been added and the text has been modified
Line 71: It would be useful to note some of the "health-promoting effects" that you refer to here. This will aid readers in understanding why you have chosen to evaluate cortisol, testosterone and Vo2max as outcomes.
A: Health-oriented aspects and references have been added
Line 103: Please confirm that participants were all aged 35-40 years. If so, the title of the article seems misleading.
A: Thank you for your comment, we have changed the title.
Line 116: It is unclear which portions or not, the control group participated in. Please make reference to a published protocol or more clearly describe the specific aspects of each group's engagements.
A: The description has been corrected.
Line 121: The abbreviation T for testosterone has already been defined. The abbreviation C for cortisol should be identified at its first use in the introduction.
A: This has been corrected
Line 143: Please identify a reference to support the statement "considered to be the most effective in the 21st century"
A: References have been added and the text has been modified
Line 152: It is unclear how you determined the pace of exercise and stage of training. Please provide clarity.
A: This has been corrected
Table 1: Is there a reason that two variations of push-ups were included? Why not an incline bench press or similar to target alternate musculature?
There is no descriptor provided for measuring the intensity of this circuit.
Please identify in the table which components relate to the control group and which relate to the intervention group.
A: The inclusion of two variations of push-ups is due to the quite different nature of the muscle work. The first exercise was performed in support on dumbbells, which were pulled towards the body when the arms were straightened, engaging the latissimus dorsi muscles, additionally increasing its range of action in the functional direction. The second type of push-ups (Exercise 9) is classic push-ups, engaging the chest muscles and static core muscles. The description in Table 1 was expanded to include that the training program applies only to the experimental group. In the control group, participants continued their previous form of activity, and, between the first and the final examinations, they followed their previous program of recreational physical activity. The intensity of the training work was supervised by a qualified personal trainer and verified by the pace of the exercises.
Where are the demographic details for each group? These should be provided given that the outcomes of interest are affected by age.
A: A table has been added to include subjects' age, weight, height, and BMI
Table 3: Given that there is a difference between groups at baseline, it is important that you provide demographic data which may aid in understanding this.
You note in the abstract that there was a "slight" change in cortisol in the intervention group. Given that p = 0.061, we should consider that there is no difference (statistically). You may later interpret this reduction as being clinically meaningful with appropriate evidence.
A: Dear Reviewer, thank you for your comment. However, we did not assume differences between groups at the beginning of the study. The study involved 30 men but only 15 of them (the experimental group) followed the original HIIT training. The entire group of 30 individuals was divided by means of number generator (randomization) into two groups: control group (15 participants), and experimental group (15 participants). In the control group, we did not change their previous form of activity, and the participants between the first and the final examination followed their previous program of recreational physical activity.
Table 5: There are no differences in Vo2max. To suggest that a "slight" increase was evident in this measure in the experimental group is inappropriate. A difference in the mean Vo2 of 80 ml/min is negligible and likely within the error of measurement for the system.
A: Changes have been made by removing the word “slight”, and, following to the comments of the second reviewer, VO2max values have been specified in ml/kg/min. The new results have been described and verified.
The first paragraph of the discussion does not seem to relate to any explanation of the results section. Please revise.
Please avoid the use of the term "insignificant". Rather, present as not significant. Whilst changes might not be statistically significant, that does not necessarily render them to be insignificant.
A: Thank you for this observation, this part has been revised and supported by more literature. This has been corrected
Line 228-230: Since the strength and endurance training used in the present study had a positive effect on testosterone levels in a group of men aged 35-40 years, improvements in their health in terms of cardiovascular disease prevention can be anticipated. Given that the majority of men in the study were already in the normal range for T, is it likely that they will experience any significant effects of this
A: The authors believe that this observation is important. Although research does not clearly indicate a positive effect of the training on the cardiovascular system (this was not studied), a review of the literature suggests a positive relationship between testosterone and the prevention of heart diseases. Our research showed that the training proposed had a positive effect on testosterone levels in the study group. Perhaps for this group of healthy men, the increase in testosterone levels will not translate into cardiovascular improvements (men were healthy), but it can be assumed that the performed training is likely to support the cardiovascular system by increasing testosterone synthesis in individuals with health problems. The paper also describes it using the conditional mode " one can hope". However, thank you for the suggestion, this will certainly be the subject of our future research.
Line 236-242: This section speaks to older men (age range not defined) and does not appear to be relevant to the current investigation.
A: This section concerns indicating some trends. They have been presented in different age ranges and an item of reference has been added. However, the literature indicates that testosterone examinations are mainly conducted in mature individuals.
Line 243-248: Why is it important to modulate the T/C ratio based on an acute response? The issue with increased cortisol or decreased T comes from chronic exposure to undesirable measures of each. Further, It would be relevant to consider that T fluctuates across the day in line with circadian rhythms. Thus, the timing of any increase in T might be more crucial than the amount in order to address daily fluctuations and the androgenic response to the exercise stimulus.
Line 243-248: Why is it important to modulate the T/C ratio based on an acute response? The issue with increased cortisol or decreased T comes from chronic exposure to undesirable measures of each. Further, It would be relevant to consider that T fluctuates across the day in line with circadian rhythms. Thus, the timing of any increase in T might be more crucial than the amount in order to address daily fluctuations and the androgenic response to the exercise stimulus.
A: A new and unproven training program proposed for middle-aged men with average fitness levels must be introduced with caution. At the beginning of a training period, monitoring physiological responses to a single session allows for adjusting the intensity of repeated efforts to individual abilities by modulating the work-to-rest ratio. All training sessions were performed in the afternoon, when circadian rhythms of T and C are not as pronounced as in the morning.
The discussion section needs to address what appears to be marginal if not inconsequential changes in Vo2max. A mean difference of less than 1 metabolic equivalent suggests that the exercise intervention had no effect on fitness.
A: Thank you for your comment. The information about VO2max and the effect of HIIT training on the level of physical fitness with has been added in the manuscript, with the example of selected indices of strength of upper body and lower and upper limbs.
Some discussion of the baseline difference in C between groups should be included here. Could this be related to demographic factors? The authors should also consider whether the lower C levels led to a floored effect for change in this outcome within the control group. Reporting of normal maximal and minimal range for this measure would assist in presenting this and its explanation.
A: No relationships were found between baseline C and demographic and anthropometric factors. The very wide range of reference values for blood cortisol levels obtained in the morning is caused, among other things, by rapid and short-lived secretion, and the maximum C concentration recorded about 30 minutes after waking up, followed by a decrease in the next 30 minutes. In all of them, the blood sampling started at 8:00 am, whereas individual wake-up time varied and was not controlled.
Given that the intervention programs were resistance training-based, it would be logical to include information on changes in strength. This may prove more useful in relation to the measurement of testosterone and cortisol and their anabolic/catabolic actions.
A: Data on changes in strength and the analysis of the relationships between these factors and hormones have been added. No such relationships were found. It seems that in this experiment, the anabolic-catabolic status was a parameter independent of strength.
Reviewer 2 Report
The research is original and is of interest to the reference sector. The paper has a good methodology and a careful analysis of the literature. I believe that the paper can be accepted in its current version.
Author Response
Dear Reviewer,
Thank you very much for your positive review of our manuscript.
Yours sincerely,
Rydzik, corresponding author
Reviewer 3 Report
My major comments are:
- Poorly described exercise intervention with no explanation of the control intervention (the pilot study keeps being mentioned but not referenced; overall the entire method section lacks more info – e.g. what VO2max protocol was used?)
- This was strength/functional program and not cardio (which is probably why no significant changes in VO2max were observed, yet the changes in VO2max were correlated with T:C ratio)
- The Intro was too long and wordy
- The aims are overstated: “develop and optimal training program." This study was not designed to answer that question. There was only 1 exercise intervention, compared to the unknown control program.
- Tables are confusing (VO2max should be reported as ml/kg/min to account for possible weight differences).
- Many other factors could have contributed to the testosterone-cortisol changes (sleep, WC, diet – only briefly mentioned) and they were not measured, reported or controlled for.
- No baseline characteristics and differences between the groups were presented.
- What are the study limitations?
- Conclusion states: "increase testosterone levels in men aged 35-40" which does not reflect the title - "men aged 30-40 yrs"
Author Response
Dear Reviewer,
Thank you very much for your time and valuable comments, which all have been considered and incorporated. The detailed list of responses is given below. We hope that the modifications and explanation will be acceptable for you.
Yours sincerely,
Rydzik, corresponding author
Poorly described exercise intervention with no explanation of the control intervention (the pilot study keeps being mentioned but not referenced; overall the entire method section lacks more info – e.g. what VO2max protocol was used?)
A: The description of the experimental training has been modified. According to us, at this point, it is more readable and clear. The control group did not follow any training program but only performed their recreational physical activity (similar to that previously performed by them every day). A more detailed description of the training protocol and the study group have been added. The procedure for VO2max measurement has also been added. The purpose of the pilot study was to verify the baseline fitness level to maintain group homogeneity. Due to the high intensity of the training, individuals who had been doing recreational training for at least 2 years were qualified to participate in the study.
This was strength/functional program and not cardio (which is probably why no significant changes in VO2max were observed, yet the changes in VO2max were correlated with T:C ratio)
The Intro was too long and wordy.
A: We agree with the Reviewer's comment. There were no significant changes in VO2max. The training was of functional and strength character. The information about VO2max and the effect of HIIT training on the level of physical fitness has been added in the manuscript, with the example of selected indices of strength of upper body and lower and upper limbs. Data on changes in strength and the analysis of the relationships between these factors and hormones have been added.
The Intro was too long and wordy
The aims are overstated: “develop and optimal training program." This study was not designed to answer that question. There was only 1 exercise intervention, compared to the unknown control program.
A: We have revised and modified the aim of the study an the Introduction section
Tables are confusing (VO2max should be reported as ml/kg/min to account for possible weight differences).
A: This has been corrected, VO2max is now given in ml/kg/min used throughout the manuscript.
Many other factors could have contributed to the testosterone-cortisol changes (sleep, WC, diet – only briefly mentioned) and they were not measured, reported or controlled for.
A: During the experiment, efforts were made to control and eliminate factors interfering e.g. diet and sleep duration. Basic information is contained in the methodology.
No baseline characteristics and differences between the groups were presented.
A: Dear Reviewer, thank you for your comment. However, we did not assume differences between groups at the beginning of the study. The study involved 30 men but only 15 of them (the experimental group) followed the original HIIT training. The entire group of 30 individuals was divided by means of number generator (randomization) into two groups: control group (15 participants), and experimental group (15 participants). In the control group, we did not change their previous form of activity, the participants between the first and the final examination followed their previous program of their own recreational physical activity (similar to that previously performed by them every day). Additionally, Table 1 provides baseline data on the study groups.
What are the study limitations?
A: This has been added in the text
Conclusion states: "increase testosterone levels in men aged 35-40" which does not reflect the title - "men aged 30-40 yrs"
A: Thank you for your comment, we have changed the title